# IMAGE-GUIDED NEURAL OBJECT RENDERING

**Justus Thies**[1]**, Michael Zollhöfer**[2]**, Christian Theobalt**[3]**, Marc Stamminger**[4]**, Matthias Nießner**[1]
[1]Technical University of Munich, [2]Stanford University, [3]Max-Planck-Institute for Informatics,
[4]University of Erlangen-Nuremberg

## ABSTRACT

We propose a learned image-guided rendering technique that combines the benefits of image-based rendering and GAN-based image synthesis. The goal of our method is to generate photo-realistic re-renderings of reconstructed objects for virtual and augmented reality applications (e.g., virtual showrooms, virtual tours & sightseeing, the digital inspection of historical artifacts). A core component of our work is the handling of view-dependent effects. Specifically, we directly train an object-specific deep neural network to synthesize the view-dependent appearance of an object. As input data we are using an RGB video of the object. This video is used to reconstruct a proxy geometry of the object via multi-view stereo. Based on this 3D proxy, the appearance of a captured view can be warped into a new target view as in classical image-based rendering. This warping assumes diffuse surfaces, in case of view-dependent effects, such as specular highlights, it leads to artifacts. To this end, we propose *EffectsNet*, a deep neural network that predicts view-dependent effects. Based on these estimations, we are able to convert observed images to diffuse images. These diffuse images can be projected into other views. In the target view, our pipeline reinserts the new view-dependent effects. To composite multiple reprojected images to a final output, we learn a composition network that outputs photo-realistic results. Using this image-guided approach, the network does not have to allocate capacity on "remembering" object appearance, instead it learns how to combine the appearance of captured images. We demonstrate the effectiveness of our approach both qualitatively and quantitatively on synthetic as well as on real data.

## 1 INTRODUCTION

In recent years, large progress has been made in 3D shape reconstruction of objects from photographs or depth streams. However, highly realistic re-rendering of such objects, e.g., in a virtual environment, is still very challenging. The reconstructed surface models and color information often exhibit inaccuracies or are comparably coarse (e.g., Izadi et al. (2011)). Many objects also exhibit strong view-dependent appearance effects, such as specularities. These effects not only frequently cause errors already during image-based shape reconstruction, but are also hard to reproduce when re-rendering an object from novel viewpoints. Static diffuse textures are frequently reconstructed for novel viewpoint synthesis, but these textures lack view-dependent appearance effects. Image-based rendering (IBR) introduced variants of view-dependent texturing that blend input images on the shape (Buehler et al., 2001; Heigl et al., 1999; Carranza et al., 2003; Zheng et al., 2009). This enables at least coarse approximation of view-dependent effects. However, these approaches often produce ghosting artifacts due to view blending on inaccurate geometry, or artifacts at occlusion boundaries. Some algorithms reduce these artifacts by combining view blending and optical flow correction (Eisemann et al., 2008; Casas et al., 2015; Du et al., 2018), or by combining view-dependent blending with view-specific geometry (Chaurasia et al., 2013; Hedman et al., 2016) or geometry with soft 3D visibility like Penner & Zhang (2017). Hedman et al. (2018) reduces these artifacts using a deep neural network which is predicting per-pixel blending weights.

In contrast, our approach explicitly handles view-dependent effects to output photo-realistic images and videos. It is a neural rendering approach that combines image-based rendering and the advances in deep learning. As input, we capture a short video of an object to reconstruct the geometry using multi-view stereo. Given this $3D$ reconstruction and the set of images of the video, we are able

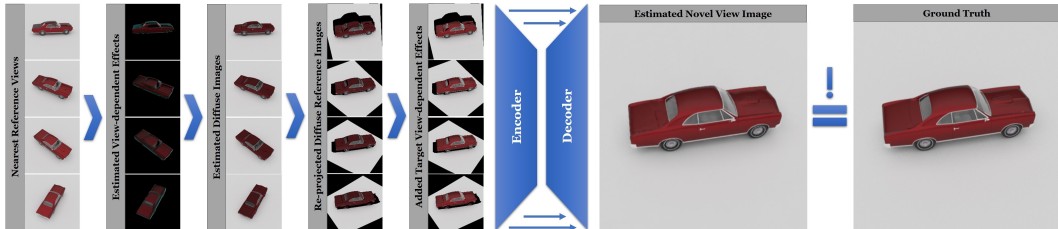

Figure 1: Overview of our image-guided rendering approach: based on the nearest neighbor views, we predict the corresponding view-dependent effects using our *EffectsNet* architecture. The view-dependent effects are subtracted from the original images to get the diffuse images that can be re-projected into the target image space. In the target image space we estimate the new view-dependent effect and add them to the warped images. An encoder-decoder network is used to blend the warped images to obtain the final output image. During training, we enforce that the output image matches the corresponding ground truth image.

to train our pipeline in a self-supervised manner. The core of our approach is a neural network called *EffectsNet* which is trained in a Siamese way to estimate view-dependent effects, for example, specular highlights or reflections. This allows us to remove view-dependent effects from the input images, resulting in images that contain view-independent appearance information of the object. This view-independent information can be projected into a novel view using the reconstructed geometry, where new view-dependent effects can be added. *CompositionNet*, a second network, composites the projected $K$ nearest neighbor images to a final output. Since *CompositionNet* is trained to generate photo-realistic output images, it is resolving reprojection errors as well as filling regions where no image content is available. We demonstrate the effectiveness of our algorithm using synthetic and real data, and compare to classical computer graphics and learned approaches.

To summarize, we propose a novel neural image-guided rendering method, a hybrid between classical image-based rendering and machine learning. The core contribution is the explicit handling of view-dependent effects in the source and the target views using *EffectsNet* that can be learned in a self-supervised fashion. The composition of the reprojected views to a final output image without the need of hand-crafted blending schemes is enabled using our network called *CompositionNet*.

## 2   RELATED WORK

**Multi-view 3D Reconstruction**   Our approach builds on a coarse geometric proxy that is obtained using multi-view 3D reconstruction based on COLMAP (Schönberger & Frahm, 2016). In the last decade, there has been a lot of progress in the field of image-based 3D reconstruction. Large-scale 3D models have been automatically obtained from images downloaded from the internet (Agarwal et al., 2011). Camera poses and intrinsic calibration parameters are estimated based on structure-from-motion (Jebara et al., 1999; Schnberger & Frahm, 2016), which can be implemented based on a global bundle adjustment step (Triggs et al., 2000). Afterwards, based on the camera poses and calibration, a dense three-dimensional pointcloud of the scene can be obtained using a multi-view stereo reconstruction approach (Seitz et al., 2006; Goesele et al., 2007; Geiger et al., 2011). Finally, a triangulated surface mesh is obtained, for example using Poisson surface reconstruction (Kazhdan et al., 2006). Even specular objects can be reconstructed (Godard et al., 2015).

**Learning-based Image Synthesis**   Deep learning methods can improve quality in many realistic image synthesis tasks. Historically, many of these approaches have been based on generator networks following an encoder-decoder architecture (Hinton & Salakhutdinov, 2006; Kingma & Welling, 2013), such as a U-Net (Ronneberger et al., 2015a) with skip connections. Very recently, adversarially trained networks (Goodfellow et al., 2014; Isola et al., 2017; Mirza & Osindero, 2014; Radford et al., 2016) have shown some of the best result quality for various image synthesis tasks. For example, generative CNN models to synthesize body appearance (Esser et al., 2018), body articulation (Chan et al., 2018), body pose and appearance (Zhu et al., 2018; Liu et al., 2018), and face rendering (Kim et al., 2018; Thies et al., 2019) have been proposed. The DeepStereo approach of Flynn et al. (2016) trains a neural network for view synthesis based on a large set of posed images. Tulsiani et al. (2018) employ view synthesis as a proxy task to learn a layered scene representation.

View synthesis can be learned directly from light field data as shown by Kalantari et al. (2016). Appearance Flow (Zhou et al., 2016) learns an image warp based on a dense flow field to map information from the input to the target view. Zhou et al. (2018) learn to extrapolate stereo views from imagery captured by a narrow-baseline stereo camera. Park et al. (2017) explicitly decouple the view synthesis problem into an image warping and inpainting task. In the results, we also show that CNNs trained for image-to-image translation (Isola et al. (2016)) could be applied to novel view synthesis, also with assistance of a shape proxy.

**Image-based Rendering**  Our approach is related to image-based rendering (IBR) algorithms that cross-project input views to the target via a geometry proxy, and blend the re-projected views (Buehler et al., 2001; Heigl et al., 1999; Carranza et al., 2003; Zheng et al., 2009). Many previous IBR approaches exhibit ghosting artifacts due to view blending on inaccurate geometry, or exhibit artifacts at occlusion boundaries. Some methods try to reduce these artifacts by combining view blending and optical flow correction (Eisemann et al., 2008; Casas et al., 2015; Du et al., 2018), by using view-specific geometry proxies (Chaurasia et al., 2013; Hedman et al., 2016), or by encoding uncertainty in geometry as soft 3D visibility (Penner & Zhang, 2017). Hedman et al. (2018) propose a hybrid approach between IBR and learning-based image synthesis. They use a CNN to learn a view blending function for image-based rendering with view-dependent shape proxies. In contrast, our learned IBR method learns to combine input views and to explicitly separate view-dependent effects which leads to better reproduction of view-dependent appearance.

**Intrinsic Decomposition**  Intrinsic decomposition tackles the ill-posed problem of splitting an image into a set of layers that correspond to physical quantities such as surface reflectance, diffuse shading, and/or specular shading. The decomposition of monocular video into reflectance and shading is classically approached based on a set of hand-crafted priors (Bonneel et al., 2014; Ye et al., 2014; Meka et al., 2016). Other approaches specifically tackle the problem of estimating (Lin et al., 2002) or removing specular highlights (Yang et al., 2015). A diffuse/specular separation can also be obtained based on a set of multi-view images captured under varying illumination (Takechi & Okabe, 2017). The learning-based approach of Wu et al. (2018) converts a set of multi-view images of a specular object into corresponding diffuse images. An extensive overview is given in the survey paper of Bonneel et al. (2017).

## 3  OVERVIEW

We propose a learning-based image-guided rendering approach that enables novel view synthesis for arbitrary objects. Input to our approach is a set of $N$ images $\mathcal{I} = \{\mathcal{I}_k\}_{k=1}^N$ of an object with constant illumination. In a preprocess, we obtain camera pose estimates and a coarse proxy geometry using the *COLMAP* structure-from-motion approach (Schönberger & Frahm (2016); Schönberger et al. (2016)). We use the reconstruction and the camera poses to render synthetic depth maps $\mathcal{D}_k$ for all input images $\mathcal{I}_k$ to obtain the training corpus $\mathcal{T} = \{(\mathcal{I}_k, \mathcal{D}_k)\}_{k=1}^N$, see Fig. 8. Based on this input, our learning-based approach generates novel views based on the stages that are depicted in Fig. 1. First, we employ a coverage-based look-up to select a small number $n \ll N$ of **fixed** views from a subset of the training corpus. In our experiments, we are using a number of $n = 20$ frames, which we call reference images. Per target view, we select $K = 4$ nearest views from these reference images. Our *EffectsNet* predicts the view-dependent effects for these views and, thus, the corresponding view-independent components can be obtained via subtraction (Sec. 5). The view-independent component is explicitly warped to the target view using geometry-guided cross-projection (Sec. 6). Next, the view-dependent effects of the target view are predicted and added on top of the warped views. Finally, our *CompositionNet* is used to optimally combine all warped views to generate the final output (Sec. 6). In the following, we discuss details, show how our approach can be trained based on our training corpus (Sec. 4), and extensively evaluate our proposed approach (see Sec. 7 and the appendix).

## 4  TRAINING DATA

Our approach is trained in an object-specific manner, from scratch each time. The training corpus $\mathcal{T} = \{(\mathcal{I}_k, \mathcal{D}_k)\}_{k=1}^N$ consists of $N$ images $\mathcal{I}_k$ and depth maps $\mathcal{D}_k$ per object with constant light.

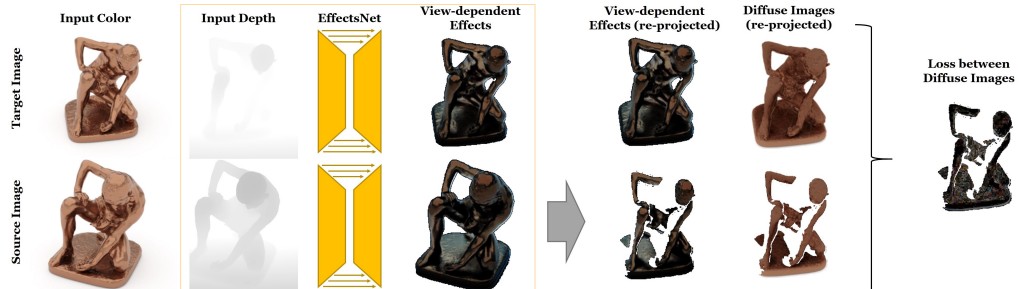

Figure 2: *EffectsNet* is trained in a self-supervised fashion. In a Siamese scheme, two random images from the training set are chosen and fed into the network to predict the view-dependent effects based on the current view and the respective depth map. After re-projecting the source image to the target image space we compute the diffuse color via subtraction. We optimize the network by minimizing the difference between the two diffuse images in the valid region.

**Synthetic Training Data**    To generate photo-realistic synthetic imagery we employ the Mitsuba Renderer (Jakob, 2010) to simulate global illumination effects. For each of the $N$ views, we ray-trace a color image $\mathcal{I}_k$ and its corresponding depth map $\mathcal{D}_k$. We extract a dense and smooth temporal camera path based on a spiral around the object. The camera is oriented towards the center of the object. All images have a resolution of $512 \times 512$ and are rendered using path tracing with 96 samples per pixel and a maximum path length of 10. The size of the training sequence is 920, the test set contains 177 images.

**Real World Training Data**    Our real world training data is captured using a *Nikon D5300* at a resolution of $1920 \times 1080$ pixels. Since we rely on a sufficiently large set of images, we record videos of the objects at a frame rate of 30Hz. Based on *COLMAP* (Schönberger & Frahm, 2016; Schönberger et al., 2016), we reconstruct the camera path and a dense point cloud. We manually isolate the target object from other reconstructed geometry and run a Poisson reconstruction (Kazhdan & Hoppe, 2013) step to extract the surface. We use this mesh to generate synthetic depth maps $\mathcal{D}_k$ corresponding to the images $\mathcal{I}_k$ (see Fig. 8). Finally, both, the color and depth images are cropped and re-scaled to a resolution of $512 \times 512$ pixels. The training corpus ranges from 1000 to 1800 frames, depending on the sequence.

## 5  *EffectsNet*

A main contribution of our work is a convolutional neural network that learns the disentanglement of view-dependent and view-independent effects in a self-supervised manner (see Fig. 2). Since our training data consists of a series of images taken from different viewing directions, assuming constant illumination, the reflected radiance of two corresponding points in two different images only differs by the view-dependent effects. Our self-supervised training procedure is based on a Siamese network that gets a pair of randomly selected images from the training set as input. The task of the network is to extract view-dependent lighting effects from an image, based on the geometric information from the proxy geometry.

**Network Inputs:**    Using a fixed projection layer, we back-project the input depth image $\mathcal{D}_i$ to world space using the intrinsic and extrinsic camera parameters that are known from the photogrammetric reconstruction. Based on this position map we generate normal maps via finite differences as well as a map of the reflected viewing directions. These inputs are inspired by the Phong illumination model (Phong, 1975) and are stacked along the dimension of the channels. Note, the network input is only dependent on the geometry and the current camera parameters, i.e., the view. Thus, it can also be applied to new target views based on the rendered depth of the proxy geometry.

**Network Architecture:**    Our network $\Phi$ is an encoder-decoder network with skip connections, similar to U-Net (Ronneberger et al., 2015b). The skip connections can directly propagate low-level features to the decoder. The encoder is based on 6 convolution layers (kernel size 4 and stride 2). The convolution layers output 32, 32, 64, 128, 256 and 512-dimensional feature maps, respectively. We use the ReLU activation function and normalize activations based on batchnorm.

The decoder mirrors the encoder. We use transposed convolutions (kernel size $4$ and stride $2$) with the same number of feature channels as in the respective encoder layer. As final layer we use a $4 \times 4$-convolution with a stride of $1$ that outputs a $3$-dimensional tensor that is fed to a Sigmoid to generate an image of the view-dependent illumination effects.

**Self-supervised Training:** Since we assume constant illumination, the diffuse light reflected by a surface point is the same in every image, thus, the appearance of a surface point only changes by the view-dependent components. We train our network in a self-supervised manner based on a Siamese network that predicts the view-dependent effects of two random views such that the difference of the diffuse aligned images is minimal (see Fig. 2). To this end, we use the re-projection ability (see Sec. 6) to align pairs of input images, from which the view-dependent effects have been removed (original image minus view-dependent effects), and train the network to minimize the resulting differences in the overlap region of the two images.

Given a randomly selected training pair $(\mathcal{I}_p, \mathcal{I}_q)$ and let $\Phi_\Theta(\mathbf{X}_t)$, $t \in \{p, q\}$ denote the output of the two Siamese towers. Then, our self-supervised loss for this training sample can be expressed as:

$$\mathcal{L}_q^p(\Theta) = \left\| M \circ \left[ \left(\mathcal{I}_p - \Phi_\Theta(\mathbf{X}_p)\right) - \mathcal{W}_q^p\left(\mathcal{I}_q - \Phi_\Theta(\mathbf{X}_q)\right) \right] \right\|_2 . \tag{1}$$

Here, $\circ$ denotes the Hadamard product, $\Theta$ are the parameters of the encoder-decoder network $\Phi$, which is shared between the two towers. $M$ is a binary mask that is set to one if a surface point is visible in both views and zero otherwise. We regularize the estimated view-dependent effects to be small w.r.t. an $\ell_1$-norm. This regularizer is weighted with $0.01$ in our experiments. The cross-projection $\mathcal{W}_q^p$ from image $p$ to image $q$ is based on the geometric proxy.

## 6 IMAGE-GUIDED RENDERING PIPELINE

To generate a novel target view, we select a subset of $K = 4$ images based on a coverage-based nearest neighbor search in the set of reference views ($n = 20$). We use *EffectsNet* to estimate the view-dependent effects of these views, to compute diffuse images. Each diffuse image is cross-projected to the target view, based on the depth maps of the proxy geometry. Since the depth map of the target view is known, we are able to predict the view-dependent effects in the target image space. After adding these new effects to the reprojected diffuse images, we give these images as input to our composition network *CompositionNet* (see Sec. 6). *CompositionNet* fuses the information of the nearest neighbor images into a single output image. In the following, we describe the coverage-based sampling and the cross-projection, and we show how to use our *EffectsNet* to achieve a robust re-projection of the view-dependent effects.

**Coverage-based View Selection** The selection of the $K$ nearest neighbor frames is based on surface coverage w.r.t. the target view. The goal is to have maximum coverage of the target view to ensure that texture information for the entire visible geometry is cross-projected. View selection is cast as an iterative process based on a greedy selection strategy that locally maximizes surface coverage. To this end, we start with $64 \times 64$ sample points on a uniform grid on the target view. In each iteration step, we search the view that has the largest overlap with the currently 'uncovered' region in the target view. We determine this view by cross-projecting the samples from the target view to the captured images, based on the reconstructed proxy geometry and camera parameters. A sample point in the target view is considered as covered, if it is also visible from the other view point, where visibility is determined based on an occlusion check. Each sample point that is covered by the finally selected view is invalidated for the next iteration steps. This procedure is repeated until the $K$ best views have been selected. To keep processing time low, we restrict this search to a small subset of the input images. This set of reference images is taken from the training corpus and contains $n = 20$ images. We chose these views also based on the coverage-based selection scheme described above. I.e., we choose the views with most (unseen) coverage among all views in an iterative manner. Note that this selection is done in a pre-processing step and is independent of the test phase.

**Proxy-based Cross-projection** We model the cross-projection $\mathcal{W}_q^p$ from image $p$ to image $q$ based on the reconstructed geometric proxy and the camera parameters. Let $\mathbf{K}_p \in \mathbb{R}^{4 \times 3}$ denote the matrix of intrinsic parameters and $\mathbf{T}_p = [\mathbf{R}_p | \mathbf{t}_p] \in \mathbb{R}^{4 \times 4}$ the matrix of extrinsic parameters of view $p$. A similar notation holds for view $q$. Then, a homogeneous 2D screen space point $\mathbf{s}_p = (u, v, d)^T \in \mathbb{R}^3$ in view $p$, with depth being $d$, can be mapped to screen space of view $q$ by: $\mathbf{s}_q = \mathcal{W}_q^p(\mathbf{s}_p)$, with

$\mathcal{W}_q^p(\mathbf{s}_p) = \mathbf{K}_q \mathbf{T}_q \mathbf{T}_p^{-1} \mathbf{K}_p^{-1} \mathbf{s}_p$. We employ this mapping to cross-project color information from a source view to a novel target view. To this end, we map every valid pixel (with a depth estimate) from the target view to the source view. The color information from the source view is sampled based on bilinear interpolation. Projected points that are occluded in the source view or are not in the view frustum are invalidated. Occlusion is determined by a depth test w.r.t. the source depth map. Applying the cross-projection to the set of all nearest neighbor images, we get multiple images that match the novel target view point.

**View-dependent Effects**   Image-based rendering methods often have problems with the re-projection of view-dependent effects (see Sec. 7). In our image-guided pipeline, we solve this problem using *EffectsNet*. Before re-projection, we estimate the view-dependent effects from the input images and subtract them. By this, view-dependent effects are excluded from warping. View-dependent effects are then re-inserted after re-projection, again using *EffectsNet* based on the target view depth map.

*CompositionNet*: **Image compositing**   The warped nearest views are fused using a deep neural network called *CompositionNet*. Similar to the *EffectsNet*, our *CompositionNet* is an encoder-decoder network with skip connections. The network input is a tensor that stacks the $K$ warped views, the corresponding warp fields as well as the target position map along the dimension of the channels and the output is a three channel RGB image. The encoder is based on 6 convolution layers (kernel size 4 and stride 2) with 64, 64, 128, 128, 256 and 256-dimensional feature maps, respectively. The activation functions are leaky ReLUs (negative slope of 0.2) in the encoder and ReLUs in the decoder. In both cases, we normalize all activations based on batchnorm. The decoder mirrors the encoder. We use transposed convolutions (kernel size 4 and stride 2) with the same number of feature channels as in the respective encoder layer. As final layer we use a $4 \times 4$-convolution with a stride of 1 and a Sigmoid activation function that outputs the final image.

We are using an $\ell_1$-loss and an additional adversarial loss to measure the difference between the predicted output images and the ground truth data. The adversarial loss is based on the conditional PatchGAN loss that is also used in Pix2Pix (Isola et al., 2016). In our experiments, we are weighting the adversarial loss with a factor of $0.01$ and the $\ell_1$-loss with a factor of $1.0$.

**Training**   Per object, both networks are trained independently using the Adam optimizer (Kingma & Ba, 2014) built into Tensorflow (Abadi et al., 2015). Each network is trained for 64 epochs with a learning rate of 0.001 and the default parameters $\beta_1 = 0.9$, $\beta_2 = 0.999$, $\epsilon = 1 \cdot e^{-8}$.

## 7   RESULTS

The main contribution of our work is to combine the benefits of IBR and 2D GANs, a hybrid that is able to generate temporally-stable view changes including view-dependent effects. We analyze our approach both qualitatively and quantitatively, and show comparisons to IBR as well as to 2D GAN-based methods. For all experiments we used $K = 4$ views per frame selected from $n = 20$ reference views. An overview of all image reconstruction errors is given in Tab. 1 in the appendix. The advantages of our approach can best be seen in the supplemental video, especially, the temporal coherence.

Using synthetic data we are quantitatively analyzing the performance of our image-based rendering approach. We refer to the Appendix A.2.1 for a detailed ablation study w.r.t. the training corpus and comparisons to classical and learned image-based rendering techniques. Since the *EffectsNet* is a core component of our algorithm, we compare our technique with and without *EffectsNet* (see Fig. 3) using synthetic data. The full pipeline results in smoother specular highlights and sharper details. On the test set the MSE without *EffectsNet* is 2.6876 versus 2.3864 with *EffectsNet*.

The following experiments are conducted on real data. Fig. 4 shows the effectiveness of our *EffectsNet* to estimate the specular effects in an image. The globe has a specular surface and reflects the ceiling lights. These specular highlights are estimated and removed from the original image of the object which results in a diffuse image of the object. In Fig. 5 we show a comparison to *Pix2Pix* trained on position maps. Similar to the synthetic experiments in the appendix, our method results in higher quality.

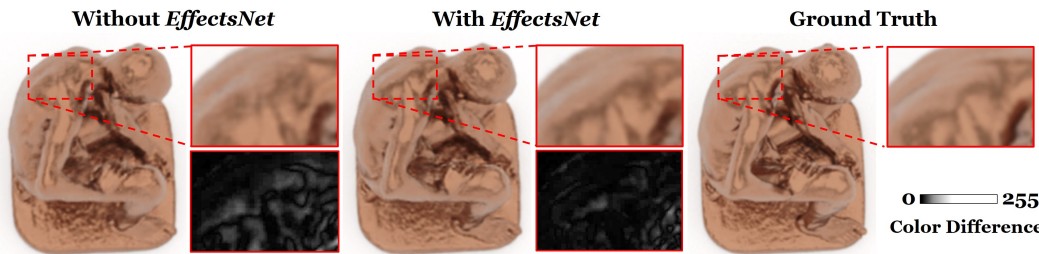

Figure 3: Ablation study w.r.t. the *EffectsNet*. Without *EffectsNet* the specular highlights are not as smooth as the ground truth. Besides, the *EffectsNet* leads to a visually consistent temporal animation of the view-dependent effects. The close-ups show the color difference w.r.t. ground truth.

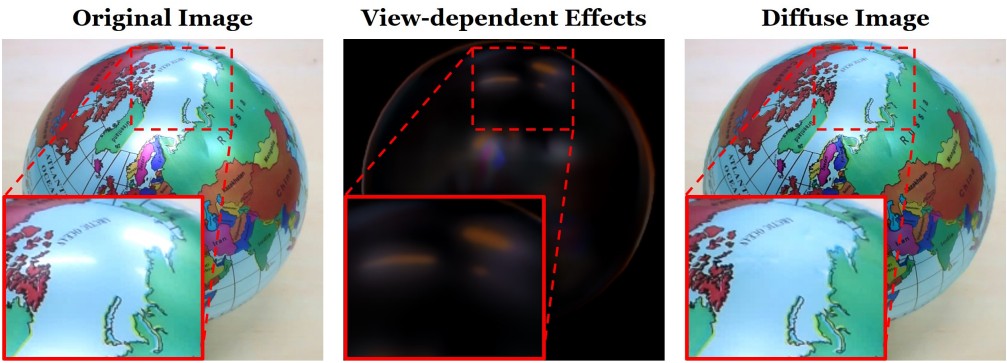

Figure 4: Prediction and removal of view-dependent effects of a highly specular real object.

We also compare to the state-of-the-art image-based rendering approach of Hedman et al. (2018). The idea of this technique is to use a neural network to predict blending weights for a image-based rendering composition like InsideOut (Hedman et al., 2016). Note that this method uses the per frame reconstructed depth maps of 20 reference frames in addition to the fused 3D mesh. As can be seen in Fig. 6, the results are of high-quality, achieving an MSE of 45.07 for DeepBlending and an MSE of 51.17 for InsideOut. Our object-specific rendering results in an error of 25.24. Both methods of Hedman et al., do not explicitly handle the correction of view-dependent effects. In contrast, our approach uses *EffectsNet* to remove the view-dependent effect in the source views (thus, enabling the projection to a different view) and to add new view-dependent effect in the target view. This can be seen in the bottom row of Fig. 6, where we computed the quotient between reconstruction and ground truth showing the shading difference.

## 8 LIMITATIONS

Our approach is trained in an object specific manner. This is a limitation of our method, but also ensures the optimal results that can be generated using our architecture. Since the multi-view stereo reconstruction of an object is an offline algorithm that takes about an hour, we think that training the object specific networks (*EffectsNet* and *CompositionNet*) that takes a similar amount of time is practically feasible. The training of these networks can be seen as reconstruction refinement that also includes the appearance of the object. At test time our approach runs at interactive rates, the inference time of *EffectsNet* is 50Hz, while *CompositionNet* runs at 10Hz on an Nvidia 1080Ti. Note that our approach fails, when the stereo reconstruction fails.

Similar to other learning based approaches, the method is relying on a reasonable large training dataset. In the appendix, we conducted an ablation study regarding the dataset size, where our approach gracefully degenerates while a pure learning-based approach shows strong artifacts.

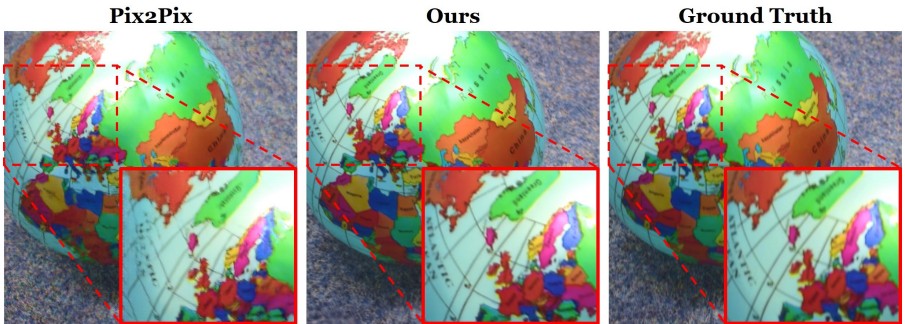

Figure 5: Comparison to *Pix2Pix* on real data. It can be seen that *Pix2Pix* can be used to synthesize novel views. The close-up shows the artifacts that occur with *Pix2Pix* and are resolved by our approach leading to higher fidelity results.

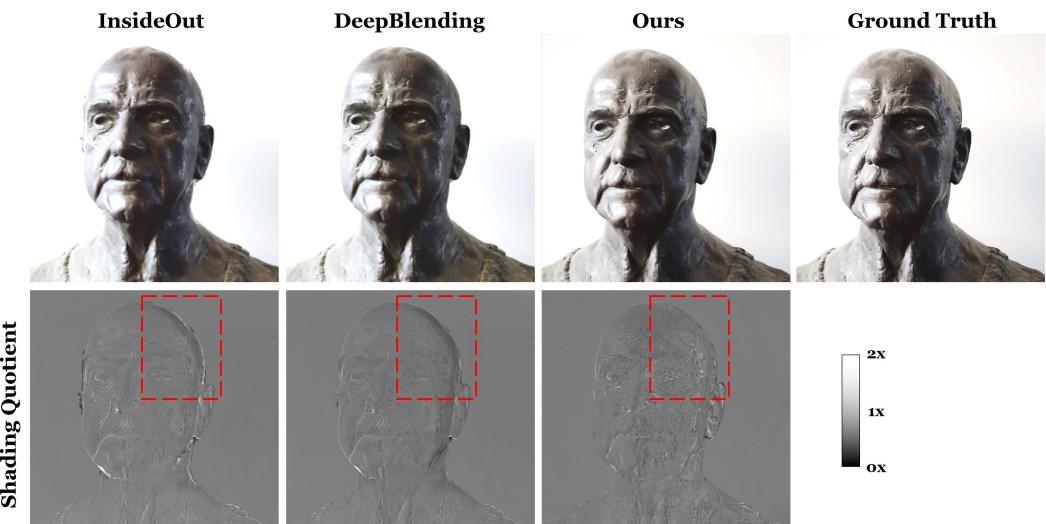

Figure 6: Comparison to the IBR method InsideOut of Hedman et al. (2016) and the learned IBR blending method DeepBlending of Hedman et al. (2018). To better show the difference in shading, we computed the quotient of the resulting image and the ground truth. A perfect reconstruction would result in a quotient of 1. As can be seen our approach leads to a more uniform error, while the methods of Hedman et al. show shading errors due to the view-dependent effects.

## 9 CONCLUSION

In this paper, we propose a novel image-guided rendering approach that outputs photo-realistic images of an object. We demonstrate the effectiveness of our method in a variety of experiments. The comparisons to competing methods show on-par or even better results, especially, in the presence of view-dependent effects that can be handled using our *EffectsNet*. We hope to inspire follow-up work in self-supervised re-rendering using deep neural networks.

## ACKNOWLEDGEMENTS

We thank Artec3D[1] for providing scanned 3D models and Angela Dai for the video voice over. This work is funded by a Google Research Grant, supported by the ERC Starting Grant Scan2CAD (804724), ERC Starting Grant CapReal (335545), and the ERC Consolidator Grant 4DRepLy (770784), the Max Planck Center for Visual Computing and Communication (MPC-VCC), a TUM-IAS Rudolf Mößbauer Fellowship (Focus Group Visual Computing), and a Google Faculty Award. In addition, this work is funded by Sony and the German Research Foundation (DFG) Grant *Making Machine Learning on Static and Dynamic 3D Data Practical*, and supported by Nvidia.

---

[1]https://www.artec3d.com/3d-models

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

# A APPENDIX

## A.1 TRAINING CORPUS

In Fig. 7, we show an overview of synthetic objects that we used to evaluate our technique. The objects differ significantly in terms of material properties and shape, ranging from nearly diffuse materials (left) to the highly specular paint of the car (right).

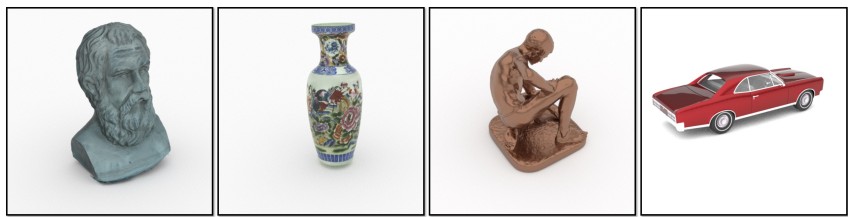

Figure 7: Renderings of our ground truth synthetic data. Based on the Mitsuba Renderer (Jakob, 2010), we generate images of various objects that significantly differ in terms of material properties and shape.

To capture real-world data we record a short video clip and use multi-view stereo to reconstruct the object as depicted in Fig. 8.

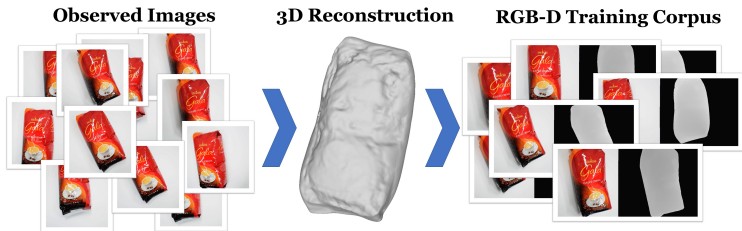

Figure 8: Based on a set of multi-view images, we reconstruct a coarse 3D model. The camera poses estimated during reconstruction and the 3D model are then used to render synthetic depth maps for the input views.

## A.2 ADDITIONAL RESULTS

An overview of all image reconstruction errors of all sequences used in this paper is given in Tab. 1. MSE values are reported w.r.t. a color range of $[0, 255]$. Note that the synthetic data contain ground truth depth maps, while for the real data, we are relying on reconstructed geometry. This is also reflected in the photo-metric error (higher error due to misalignment).

|  | Sequence | N-IBR | IBR (Debevec et al. (1998)) | Pix2Pix (Isola et al. (2016)) | **Ours** |
|---|---|---|---|---|---|
| Synthetic | Fig. 1, Car | 72.16 | 39.90 | 12.62 | **3.61** |
| | Fig. 3, Statue | 44.27 | 25.01 | 5.40 | **2.38** |
| | Fig. 11, Vase | 38.93 | 17.31 | 18.66 | **1.12** |
| | Fig. 12, Bust | 35.58 | 20.45 | 4.43 | **1.52** |
| Real | Fig. 4, Globe | 152.21 | 81.06 | 154.29 | **30.38** |
| | Fig. 14, Shoe | 98.89 | 59.47 | 116.52 | **56.08** |
| | Fig. 6, Bust | 397.25 | 72.90 | 45.23 | **25.24** |

Table 1: MSE of photometric re-rendering error of the test sequences (colors in [0-255]). Pix2Pix is trained on world space position maps. N-IBR is the naïve blending approach that gets our nearest neighbors as input.

### A.2.1 ADDITIONAL EXPERIMENTS ON SYNTHETIC DATA

Using synthetic data we are quantitatively analyzing the performance of our image-based rendering approach.

***EffectsNet*** In Fig. 9, we show a qualitative comparison of our predicted diffuse texture to the ground truth. The figure shows the results for a *Phong* rendering sequence. As can be seen, the estimated diffuse image is close to the ground truth.

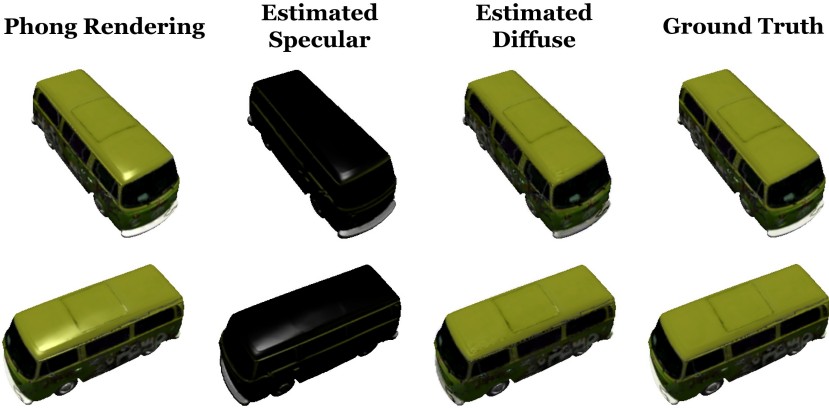

Figure 9: Comparison of the estimated diffuse images based on *EffectsNet* and the ground truth renderings. The input data has been synthesized by a standard *Phong* renderer written in *DirectX*. The training set contained 4900 images.

**Comparison to Image-based Rendering** We compare our method to two baseline image-based rendering approaches. A *naïve IBR* method that uses the nearest neighbor of our method and computes a per pixel average of the re-projected views, and the IBR method of Debevec et al. (1998) that uses all reference views and a per triangle view selection. In contrast to our method, the classical IBR techniques are not reproducing view-dependent effects as realistically and smoothly which can be seen in Fig. 10. As can be seen the naïve IBR method also suffers from occluded regions. Our method is able to in-paint these regions.

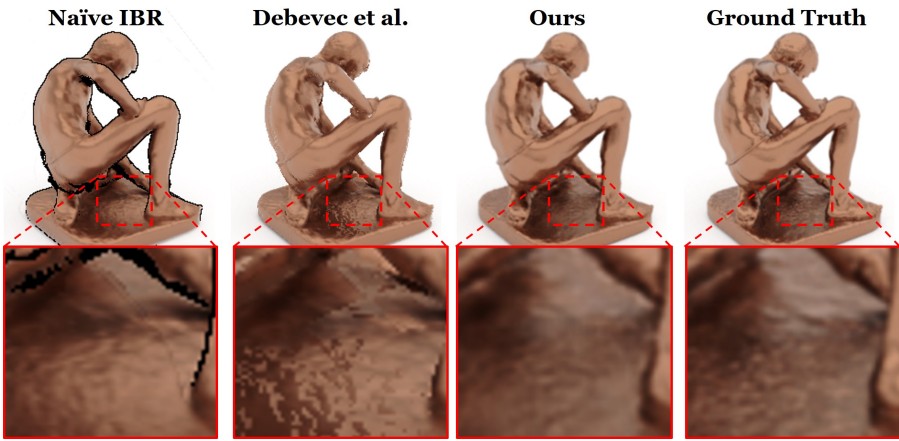

Figure 10: Comparison of our neural object rendering approach to IBR baselines. The naïve IBR method uses the same four selected images as our approach as input and computes a pixel-wise average color. The method of Debevec et al. (1998) uses all reference views ($n = 20$) and a per triangle view selection. The training set contained 1000 images.

**Comparison to Learned Image Synthesis** To demonstrate the advantage of our approach, we also compare to an image-to-image translation baseline (*Pix2Pix* (Isola et al., 2016)). *Pix2Pix* is trained to translate position images into color images of the target object. While it is not designed for this specific task, we want to show that it is in fact able to produce individual images that look realistic (Fig. 11); however, it is unable to produce a temporally-coherent video. On our test set with 190 images, our method has a MSE of 1.12 while *Pix2Pix* results in a higher MSE of 18.66. *Pix2Pix* trained on pure depth maps from our training set results in a MSE of 36.63 since the input does not explicitly contain view information.

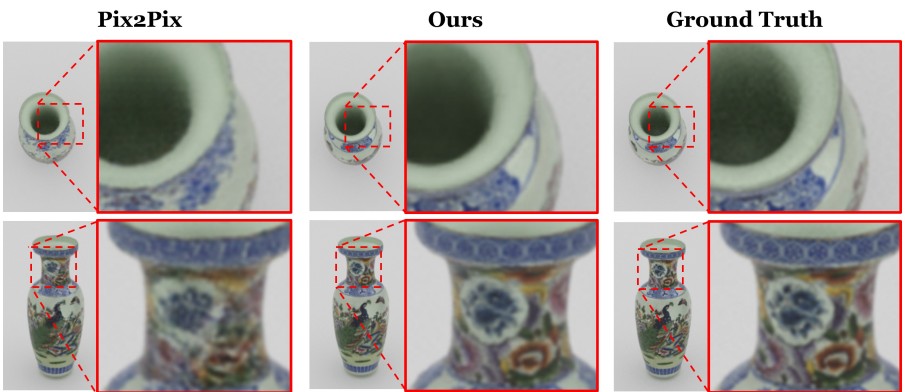

Figure 11: In comparison to *Pix2Pix* with position maps as input, we can see that our technique is able to generate images with correct detail as well as without blur artifacts. Both methods are trained on a dataset of 920 images.

**Evaluation of Training Corpus Size** In Fig. 12 we show the influence of the training corpus size on the quality of the results. While our method handles the reduction of the training data size well, the performance of *Pix2Pix* drastically decreases leading to a significantly higher MSE. When comparing these results to the results in Fig 11 it becomes evident that *Pix2Pix* has a significantly lower error on the bust sequence than on the vase sequence. The vase has much more details than the bust and, thus, is harder to reproduce.

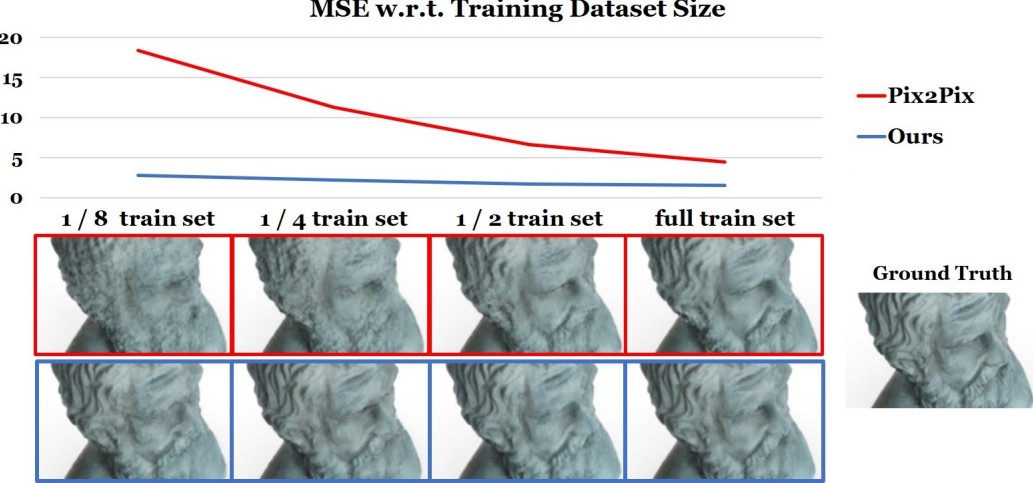

Figure 12: In this graph we compare the influence of the training corpus size on the MSE for our approach and *Pix2Pix* trained on position maps. The full dataset contains 920 images. We gradually half the size of the training set. As can be seen, the performance of our approaches degrades more gracefully than *Pix2Pix*.

### A.2.2 Additional Experiments on Real Data

**Comparison to Texture-based Rendering** Nowadays, most reconstruction frameworks like COLMAP (Schönberger & Frahm, 2016), KinectFusion (Izadi et al., 2011), or VoxelHashing (Nießner et al., 2013) output a mesh with per-vertex colors or with a texture, which is the de facto standard in computer graphics. Fig. 13 shows a side-by-side comparison of our method and the rendering using per-vertex colors as well as using a static texture. Since both the vertex colors as well as the texture are static, these approaches are not able to capture the view-dependent effects. Thus, view-dependent effects are baked into the vertex colors or texture and stay fixed (seen close-ups in Fig. 13).

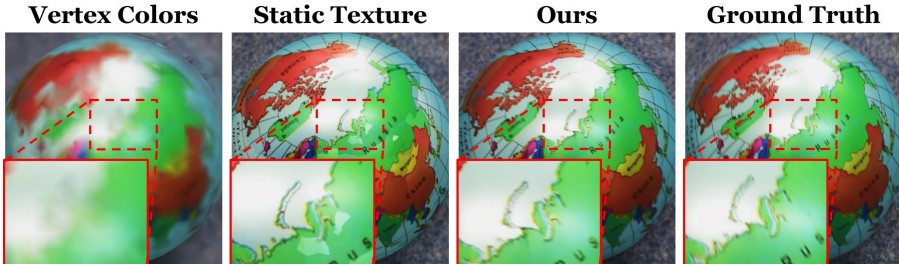

Figure 13: Image synthesis on real data in comparison to classical computer graphics rendering approaches. From left to right: Poisson reconstructed mesh with per-vertex colors, texture-based rendering, our results and the ground truth. Every texel of the texture is a cosine-weighted sum of the data of four views where the normal points towards the camera the most.

**Comparison to Image-based Rendering** Fig. 14 shows a comparison of our method to image-based rendering. The IBR method of Debevec et al. (1998) uses a per triangle based view selection which leads to artifacts, especially, in regions with specular reflections. Our method is able to reproduce these specular effects. Note, you can also see that our result is sharper than the ground truth (motion blur), because the network reproduces the appearance of the training corpus and most images do not contain motion blur.

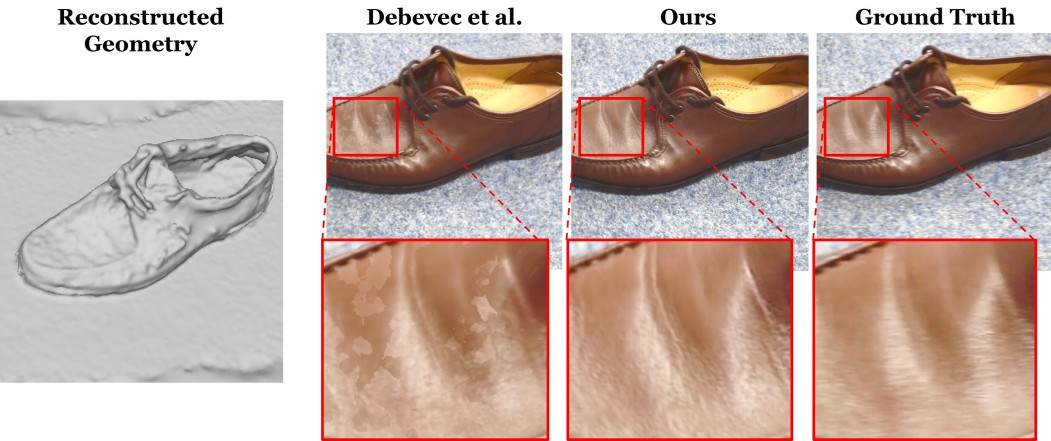

Figure 14: Image synthesis on real data: we show a comparison to the IBR technique of Debevec et al. (1998). From left to right: reconstructed geometry of the object, result of IBR, our result, and the ground truth.

