# OpenReview forum: "Image-guided Neural Object Rendering"
_ICLR.cc/2020/Conference — Accept (Poster)_

### Official Review · AnonReviewer1 · 2019-10-23
**Official Blind Review #1**

**Rating:** 6

**Review:**

The submission proposes a method to perform neural rendering. From a set of images taken of a static object under constant illumination, the proposed method first selects the four viewpoints nearest to the requested novel viewpoint. Then, the images corresponding to those viewpoints are blended together using an encoder-decoder network to produce the novel view image. The key contribution of the submission over previous work is the handling of view-dependent effects such as specular highlights. Those view-dependent effects are first removed from the nearest retrieved images using the proposed EffectsNet. The resulting estimated diffuse images are then re-projected to the target viewpoint and view-dependent effects from this viewpoint are added before blending the images together.

The proposed method improves the robustness of existing neural rendering methods to materials that are not roughly diffuse.

The process to select the 20 reference images is based on a coverage scheme that is never presented, hindering reproducibility. Would it be possible to describe this scheme? Similarly, the hyperparameters used for the adversarial loss are not explicitly stated, are they exactly the same as the cited Pix2Pix? Would the source code be shared publicly?

I would have appreciated an ablative experiment where the proposed pipeline is kept as-is, but a state-of-the-art intrinsic decomposition technique (as discussed in sec. 2) was applied instead of EffectsNet.

I would recommend using subsections to prevent confusion in references (for example, sec. 6, p. 5 referring to sec. 6).

In fig. 6, the quotient image is hard to interpret as it is not linear. A colormap representing the percentage of error wrt. the ground truth might be easier to interpret.

EffectsNet is used for both removal and addition of view-dependent effects, which makes part of the paper confusing. Maybe adding the mathematical symbols of eq. 1 to fig. 2 might help the reader to understand the training steps?

The impact of the regularizer weight of 0.01 (p. 5) is not discussed. I suspect this value might be important, as too much regularization might give underestimated view-dependent effects and too little might provide strong and incoherent effects. An analysis of this hyperparameter would be welcome.

In my opinion, the proposed abstract is slightly hard to read, maybe it would benefit from being shorter?

Minor details
- p. 3 “An extensive overview is give[n] in [...]”
- p. 8 I believe the “z” of the unit “Hz” is wrongly stylized as a mathematical variable.


**Experience Assessment:**

I have read many papers in this area.

**Review Assessment: Checking Correctness Of Derivations And Theory:**

I assessed the sensibility of the derivations and theory.

**Review Assessment: Checking Correctness Of Experiments:**

I assessed the sensibility of the experiments.

**Review Assessment: Thoroughness In Paper Reading:**

I read the paper at least twice and used my best judgement in assessing the paper.

---

> ### Public Comment · ~Jack_William1 · 2022-10-22
> **Where can I buy English essays?**
>
> It's always best to hire a professional essay writer who will take the time to understand your specific needs and write a paper that is tailored to your requirements. You can find reputable essay writing services online or by doing a search for write my english essay for me https://www.assignmentholic.co.uk/english-essay-help/ in your local area.

---

### Official Review · AnonReviewer2 · 2019-10-23
**Official Blind Review #2**

**Rating:** 8

**Review:**

The paper presents a method for rendering of reconstructed objects. At its core, it works by distinguish between diffuse images and "view-dependent effects" (in essence, every element related to the scene instead of the object like specularities). The approach leverages two distinct neural networks. First, EffectsNet, which infers the view-dependent effects from an image and their depth map. Second, CompositionNet, which learns to combine multiple warped view into the final image and to remove the related artifacts. Results show improvements with respect to previous methods, especially for temporal coherence and dealing with a low number of images.

The proposed approach is well described, and does seem to fix a few issues with current methods. The comparisons are extensive and comprehensive for what I can tell. The results on static images are not significantly better than previous work, however, and the need to train the whole pipeline for every object is quite a limitation. Apart these considerations, I do not have much complaints about this work. The few others I came up with are provided below.

I am not completely sure here, but it seems like CompositionNet could be used with other approaches than the one described in the paper, in particular with IBR. How would it improve the performance of previous approaches, especially regarding temporal coherence?

For the EffectsNet training, we have to assume that two random images from the training set can be reprojected onto one another without loosing too much of the surface. If the views provided by these images are too different, than the number of pixels actually providing a loss to the network will be very low, which would be detrimental to the learning process. Could you comment on that? Is there a quick fix to this issue?

One thing that was not discuss was about the effects integrating a neural network into the rendering pipeline have on the output resolution. Classical methods are not really limited on this regard, but neural networks cannot easily provide high resolution outputs. I understand that EffectsNet is not directly used as image (only to subtract the "view-dependent" parts), but could there be issues if we use a resolution of, say, 2048x2048 instead of 512x512? Or even higher? Given the aim of the paper at targeting high quality rendering, such consideration has important practical consequences.

In summary, this paper does a reasonable job at improving the current IBR methods. The idea of making the network learns the scene effects instead of the whole object appearance is sound and could probably be translated to other related problems.

**Experience Assessment:**

I have published one or two papers in this area.

**Review Assessment: Checking Correctness Of Derivations And Theory:**

N/A

**Review Assessment: Checking Correctness Of Experiments:**

I assessed the sensibility of the experiments.

**Review Assessment: Thoroughness In Paper Reading:**

I read the paper at least twice and used my best judgement in assessing the paper.

---

### Official Review · AnonReviewer4 · 2019-11-03
**Official Blind Review #4**

**Rating:** 3

**Review:**

Authors propose a novel neural image-guided rendering method, a hybrid between classical image-based rendering and machine learning. They learn EffectsNet in a self-supervised manner to capture the view-dependent component using a reprojection loss. To render a novel view they subtract a view-dependent component from K closest train views, reproject the diffuse component to the novel view, add view-dependent component, use a CompositionNet to render a final image. They use both synthetic and real data for the experiments and compare to state-of-the-art image-based rendering methods of Hedman et al.


I think the method discussed in the paper is a nice contribution to the neural rendering area. The paper is written clearly and the provided amount of technical details is enough to reproduce the method.


I only have some concerns about the experiments.

1) The method relies on the depth maps and the main contribution of the paper is related to the rendering of view-dependent effects. However, in realistic scenarios, the presence of the view-dependent effects leads to severe degradation of photogrammetry pipeline: imprecise camera poses and geometry. To my understating, imprecise geometry and camera poses should affect the method heavily. I think, the paper lacks a thorough evaluation on more realistic data, having only a single example in Fig. 14.

2) Authors compare to DeepBlending of Hedman et al. and although the proposed model achieves the best quantitative result, to my mind, the qualitative difference is not very noticeable, while it is known that MSE error does not correlate with human perception good enough. To justify the presented method it would be great to see experiments with a more clear difference to Hedman et al. baseline.

3) Authors of Pix2Pix (Isola et al., 2016) do not provide experimental results on novel view-point synthesis. Although it is completely fine to refer to Pix2Pix as to the closest and simplest neural-based baseline I do not find it convincing to compare the proposed method to Pix2Pix without any other baselines as in the figures 5, 11, 12. Pix2pix had never been claimed to be developed or validated for this particular problem so it is a kind of "artificial" baseline. Moreover, I believe a stronger "artificial" baseline can be easily constructed. In particular, it is hard to believe that for the "Vase" scene, which has almost no view-dependent effects, a network that takes a world space position map cannot learn a direct mapping from (x,y,z) to (r, g, b). That is why, I believe, a stronger "artificial" baseline could be easily constructed by some kind of tweaking of Pix2Pix, say, turning off GAN or changing neural networks' architecture slightly.

Considering the issues mentioned above my I tend to recommend this paper for rejection.


Questions:

1) Video shows examples of novel view synthesis for a predefined camera trajectory. How this trajectory is created for both synthetic and real objects? How does it differ from the train trajectory?


Some minor comments:

1) In Fig. 2 it reads "Input color", whereas the RGB color image is not used as input to EffectsNet. It confuses on the first read. In "... based on the current view and the respective depth map", it is also easy to think that "current view" refers to RGB image.

2) The model from Figure 8 is never used.



**Experience Assessment:**

I have published one or two papers in this area.

**Review Assessment: Checking Correctness Of Derivations And Theory:**

N/A

**Review Assessment: Checking Correctness Of Experiments:**

I carefully checked the experiments.

**Review Assessment: Thoroughness In Paper Reading:**

I read the paper thoroughly.

---

### Official Review · AnonReviewer3 · 2019-11-04
**Official Blind Review #3**

**Rating:** 6

**Review:**

This paper studies the problem of re-rendering the appearance of an object from novel viewpoints, given reference images of the same object as input. Multi-view reconstruction and camera poses are used to generate depth images to be used as additional input to the network. The authors propose an EffectsNet architecture to derive view-independent Lambertian reflectance maps and train the network in a self-supervised way through re-projection. From the target viewpoint, the image is re-textured by a blending network and view-dependent lighting effects are added back, to generate a novel-view rendering.

Application may be limited due to the fact that the network is re-trained from scratch for each object instance. Another weakness is insufficient evaluation against other deep-learning based IBR approaches. It is nice to see comparison with Hedman et al. (2016 and 2018) but I feel that details are missing. Figure 6 and L3-4 of page 8 claim a better MSE and qualitative results for a single example, but the difference is not obvious and to validate the claim that the proposed approach "leads to superior reproduction of view-dependent appearance" (Page 3 under Image-based Rendering), I think a more systematic evaluation on multiple objects and views is needed.

Despite the weaknesses, I am leaning towards Weak Accept because I think the benefits (requiring less training examples and being able to generate a more temporally coherent images) are convincing. Using surface reflectance decomposition to reduce photometric inconsistency in projection-based self-supervision is also novel for this task. But I think a more thorough comparison with Hedman et al. would be crucial in justifying the design choices.

One potential improvement for future work is that it may be possible to compute confidence maps in the diffuse image estimation step and use that to improve blending of the warped images.

**Experience Assessment:**

I have read many papers in this area.

**Review Assessment: Checking Correctness Of Derivations And Theory:**

I assessed the sensibility of the derivations and theory.

**Review Assessment: Checking Correctness Of Experiments:**

I assessed the sensibility of the experiments.

**Review Assessment: Thoroughness In Paper Reading:**

I made a quick assessment of this paper.

---

### Author Response · Authors · 2019-11-15
**Reply to the reviewers**

Dear Reviewers,
Thank you for your detailed reviews!
All reviewers are positive regarding our proposed method.

Some concerns regarding the experiments were raised that we want to clarify in a minor revision. Specifically, we will improve the presentation of the comparison to DeepBlending of Hedman et al. to highlight the differences (we will also show the shading quotient in the video).
We agree that Pix2Pix is not designed for novel view-point synthesis (we will clarify that more prominently in the paper). Nevertheless, it is closely related to our approach and also outputs reasonable results with default settings (see video, e.g., real sequences).
In the video, we show three sequences on real data, but as stated by reviewer #4 our approach relies on correctly reconstructed camera poses / geometry. Thus, when the stereo reconstruction fails, our algorithm also fails; we will clarify this limitation.

Besides the experiments, we will also include more information about the selection procedure of the reference frames and improve the notation for the EffectsNet section as well as the training of the network. We will also rewrite the abstract for better readability.

Q&A:
- How does it [the test trajectory] differ from the train trajectory?
For the real sequences we record a long sequence with a random path around the object. We take the first part of the video for training and the second part for testing.
For synthetic scenes we randomly sample from a sphere for training and use a smooth spiral trajectory for testing (both the sphere and the spiral have the same radius)

- [EffectsNet training] If the views provided by these images are too different, than the number of pixels actually providing a loss to the network will be very low, which would be detrimental to the learning process.
We normalize the loss by the number of ‘shared pixels’.

- [Output resolution] Could there be issues if we use a resolution of, say, 2048x2048 instead of 512x512?
To achieve a similarly large receptive field w.r.t. the actual captured scene, one would need to increase the network depth, or use a hierarchical approach (coarse to fine).


Thank you for helping us to improve our submission!
Best,
The authors.

---

### Public Comment · ~Jack_Ali1 · 2023-05-06
**GPS**

GPS is also used for targeting in the military. It allows for the precise targeting of enemy forces and infrastructure, which can reduce collateral damage and civilian casualties. https://routefinderhq.com/best-gps-for-boat/ This is particularly important in urban combat situations, where accuracy is crucial.

---

> ### Public Comment · ~Fahim_Ferdoush1 · 2023-07-17
> **Thanks**
>
> Great info. Thanks for sharing. If You looking for Any WordPress service or wordpress Expert help here is a source- https://prowordpressexpert.com/

---

### Decision · Program_Chairs · 2019-12-19

**Decision:**

Accept (Poster)

**Comment:**

The paper presents a new variation of neural (re) rendering of objects, that uses a set of two deep ConvNets to model non-Lambertian effects associated with an object. The paper has received mostly positive reviews. The reviewers agree that the contribution is well-described, valid and valuable. The method is validated against strong baselines including Hedman et al., though Reviewer4 rightfully points out that the comparison might have been more thorough.

One additional concern not raised by the reviewers is the lack of comparison with [Thies et al. 2019], which is briefly mentioned but not discussed. The authors are encouraged to provide a corresponding comparison (as well as additional comparisons with Hedman et al) and discuss pros and cons w.r.t. [Thies et al] in the final version.